# Knowledge and Practices of Cypriot Bovine Farmers towards Effective and Safe Manure Management

**DOI:** 10.3390/vetsci10040293

**Published:** 2023-04-14

**Authors:** Soteris Christophe, Kristina Pentieva, George Botsaris

**Affiliations:** 1Cyprus Veterinary Services, Athalassas Av., Nicosia 1417, Cyprus; 2School of Biomedical Sciences, Ulster University, Coleraine BT52 1SA, UK; 3Department of Agricultural Sciences, Biotechnology and Food Science, Cyprus University of Technology, Limassol 3603, Cyprus

**Keywords:** bovine farmers, manure management, knowledge, practices, manure treatment

## Abstract

**Simple Summary:**

Manure generated in bovine farms may pose human and animal health risks. If not managed and used properly, these risks may be spread to other areas. The risks can be minimised when farmers are aware of them and apply optimum practices at their farms, at storage, processing, and when manure is applied to land as a fertiliser. This research aims to evaluate the Cypriot bovine farmers’ knowledge about the risks from manure and the practices currently applied for manure management through a questionnaire survey. The results indicated some gaps in farmers’ knowledge and some deficiencies in employing optimal management practices. The education level and the farming purpose were identified as determinants of the level of farmers’ knowledge. The conclusion was that farmers’ knowledge must be reinforced through specialised training to ensure proper manure management. Although the current practices partially decrease manure pathogens, interventions to promote the use of more effective treatment methods, such as biogas transformation and composting, would be beneficial. The results could help Cypriot and other competent authorities to develop an action plan to control the health risks by educating farmers, promoting the use of more efficient management methods and developing new legislation.

**Abstract:**

Manure from bovine farms is commonly used as an organic fertiliser. However, if not properly managed, it can spread significant biological and chemical hazards, threatening both human and animal health. The effectiveness of risk control hugely relies on farmers’ knowledge regarding safe manure management and on the application of suitable management practices. This study aims to evaluate the knowledge and practices of Cypriot bovine farmers towards safer manure management, from its generation to its final use, in line with the One Health approach. Factors affecting farmers’ knowledge and applied practices are also investigated through a questionnaire survey. The questionnaire was developed and sent to all eligible bovine farmers in Cyprus (n = 353), and 30% (n = 105) of them returned the completed questionnaire. Results revealed there are some gaps in farmers’ knowledge. The use of manure for fertilising crops dominated. Only half of the farmers stored manure in appropriate facilities, with 28.5% of them using a dedicated area with cement floors and 21.5% utilising leakproof tanks. The majority (65.7%) stored manure for more than three months before its use as a fertiliser in a dried form. In multiple regression analysis, education level and farming purpose were significant determinants of farmer knowledge. In conclusion, Cypriot farmers’ knowledge must be reinforced to ensure proper manure management. The results highlight the importance of providing relevant training to farmers. Although the current practices partially decrease manure pathogens, interventions to promote the use of more effective treatment methods, such as biogas transformation and composting, would be beneficial.

## 1. Introduction

Manure is among the most hazardous livestock farming by-products for human and animal health as it contains a substantial amount of pathogens [1] that present severe biological and chemical hazards [2]. In the EU and UK together, 1.4 billion tonnes of manure are generated each year, of which 75% is of bovine origin [3]. Because of the limited storage and processing facilities in farms, about 92% of the manure is applied directly to land unprocessed [3]. Cyprus is facing similar problems—in its five districts, the bovine population in 2021 was approximately 84,000 animals, and it is estimated that they generated about 1,125,000 tonnes of manure, representing more than half of total manure quantities in Cyprus [3]. There are no official data for the final disposal or use of this manure, but there is evidence that the most common practice is the storage in deep pits and anaerobic lagoons before land application [4].

Manure may harbour a variety of pathogens; bacteria, including *Salmonella* spp., *Campylobacter* spp. and *E. coli*; viruses, including apthoviruses, enteric viruses, and myxoviruses; and various parasites, mycoses and yeasts [5,6,7,8,9,10]. It has been implicated in several incidences of spreading diseases such as *E. coli* (0157:H7) diarrhoea, Cryptosporidiosis, and Salmonellosis [11,12,13]. Additionally, chemical substances that are routinely used or are unintentionally present in farming may be traced in manure. Consequently, it may contain antibiotic residues [14,15,16,17,18] and antibiotic-resistant bacteria [19,20,21] or other contaminants, including dioxins [22], heavy metals [23], illegal drugs [24] and hormones [25,26,27,28,29]. Improper manure storage, handling, and management may pollute the environment and expose humans and animals to the aforementioned hazards [9] via contaminated food, water, dust, wild animals, pests and others [5,10,30,31]. Traditionally, manure was used for fertilising crops and improving soil quality [9,32]. The intensification of breeding systems has led to excess manure production, which is above the local needs. This necessitates the storage of manure on the farm and the dispatch to other regions for application [33], which in turn, promotes the spreading of pathogens [9].

Risks to human and animal health can be mitigated by applying good management practices and taking prevention measures in three stages [10]. The first is the application of good farming practices to reduce the initial manure’s pathogen load and hazardous residues. These include pest control programmes, good husbandry practices, maintaining an ideal level of animal welfare, the prudent use of veterinary medicines, a vaccination programme, parasite control and others [34,35,36]. The second level is based on chemical, physical, or biological manure decontamination methods. These reduce health risks [9] despite being developed for other purposes such as volume reduction and profitability [37]. Among manure decontamination methods, the pasteurisation at 70 °C for 1 h following biogas transformation or aerobic composting is distinguished for its effectiveness [32,37,38,39,40]. Finally, it is important to follow the best practices during manure’s use as a fertiliser, including prompt incorporation into land and avoiding sloping grounds to prevent run-off after intense precipitation [10,41,42,43,44,45]. The effect of the sun’s UV radiation, the soil bacteria and desiccation [10,46] play a key role in reducing pathogens.

Historically, manure is used directly on land or after a certain storage period. Consequently, there is minimal pathogen reduction, if any. The European Parliament and European Council Regulation (EC) No. 1069/2009 and Commission Regulation (EU) No. 142/2011, commonly referred to as the animal by-products regulations, allow for the direct application of manure to land without any prior treatment within the territory of the member state if the competent authority approves such use. Therefore, the application of unprocessed manure on land is a common practice among EU member states. Although Regulation (EU) 2019/1009 allows the use of animal by-products, including manure in fertilising products provided that are to be treated to a specified level of safety defined in the animal by-products regulations, it does not exclude the use of unprocessed manure within the member state of production if not used to produce fertilising products. The national legislation of Cyprus for manure management focuses on environmental issues. Specifically, it consists of two acts issued based on the Law for the Control of Water Pollution 106(I)/2002, namely the Regulatory-Administrative Act 263/2007 for “Good Agricultural Practices” and the Act 433/2006 for the General Rules for the Disposal of Waste from Bovine Farms” aiming against the nitrate pollution of underground waters. The acts set rules for storing and managing manure in the farm and its use on land as a fertiliser. Among others, they recommend three-month storage of the manure before application on land, cement floors for storage of manure in the farm with a slope towards a leakproof tank for separation of the liquid fraction and quicker air drying of the manure. Consequently, the risk of spreading live pathogens and other hazardous agents from the use of manure is expected to remain a significant challenge.

While health risks arising from the improper use or disposal of manure and incidences of zoonotic outbreaks related to contaminated manure are being studied intensively [7,9,10], the safe management of manure remains a significant problem [47,48] since data about farmers’ knowledge and applied practices for safe manure management are scarce. To the best of our knowledge, so far, there is no research investigating the knowledge of farmers regarding human and animal health risks associated with improper manure management and current management practices in Cyprus and other European countries, which is a serious gap in this field. Considering the significant contribution of bovine manure the total manure production, this study aims to evaluate bovine farmers’ knowledge about public and animal health risks associated with manure generated in their farms and to identify current practices applied for manure management in Cyprus. Furthermore, it aims to investigate the demographic and other determinants that affect the farmers’ knowledge and applied practices, so appropriate interventions can be implemented to improve the safety of manure management following a holistic approach to the issue, starting from breeding practices to the final use, contributing towards the One Health perspective.

## 2. Materials and Methods

### 2.1. Study Population

To evaluate the knowledge and practices of bovine farmers towards effective and safe manure management, a questionnaire survey was conducted. The study population included all farmers of bovine animals who have active animal holdings in the areas under the effective control of the Government of the Republic of Cyprus. The necessary information, including farmers’ contact details, was obtained from the database of the Veterinary Services of Cyprus. Bovines were kept in 384 farms (average farm size: 220 animals), belonging to 374 physical or legal persons [49]. In 2021, 21 farms were focusing on beef production exclusively, whereas all the rest were dairy farms or dairy mixed with beef. Dairy breeds, mainly Holstein-Friesians, represent more than 97% of the population. A small number of animals belong to Jersey, Limousin, Black Angus, the local red breed, Swiss Brown, Charolais and other breeds. The animals are kept confined in farms except for about 200 animals belonging to the local breed in the pastureland of Akrotiri of Lemesos District [50]. From the 374 farmers listed in the database, 21 organisations, research institutes and farmers currently with no animals were excluded. All other farmers in the database, 353, were eligible for participation and invited to contribute to the research.

### 2.2. Questionnaire Development and Administration

The questionnaire was developed and designed de novo to evaluate the knowledge and practices followed during the three critical stages for controlling the risks from manure [51]; in particular, to evaluate the application of good farming practices, the use of decontamination methods before manure use, and the practices followed when it is applied to land. It comprised three sections: Demographics (farmer’s age, years in the profession, education and farm location, size and farming purpose), Assessment of Knowledge, and Identification of Current Practices/Approaches (Appendix A). The questionnaire was written in Greek, which is the language spoken in Cyprus. It included a combination of question types; for example, dichotomous questions (yes/no; right/wrong); Likert scale questions (“Not at all important”, “Slightly important”, “Important” and “very important”); and free numerical responses (for example “Please state your age”). For the assessment of the knowledge, a scoring system was developed analogous to that of Tierney et al. [52], who allocated a point for each correct answer. That part of the questionnaire consisted of 4 main questions, each having several sub-questions, for a total of 31 questions. The questions were related to safe manure management practices, diseases and other hazards transmitted through manure, treatment methods to reduce the risks, and general right or wrong statements. The highest score a responder could get was 31 points (6 points for question 1, 11 for question 2, 8 for question 3 and 6 for question 4).

The questionnaire was piloted on ten independent persons from diverse backgrounds but relevant to the field to check it for clarity, simplicity and precision [53]. Based on their feedback, the questionnaire was revised accordingly and finalised. The estimated completion time was five to ten minutes. After the ethical approval was granted, a package containing an invitation letter providing information about the study, together with the questionnaire and a stamped (prepaid), self-addressed envelope for returning the completed questionnaire to the researcher, was sent to farmers. The invitation letter explained the study’s scope, participants’ rights and confidentiality issues, researcher’s contact details, instructions on how to return the completed questionnaire and other relevant details. To ensure the anonymity of the survey, the questionnaire did not include any questions or other elements that could lead to the identification of responders. The questionnaire distribution started in May 2022, and responses were received until mid-July 2022. Some of the envelopes were distributed by a farmers’ organisation to its members and some by personnel of the Veterinary Services while visiting farms for their routine work. Care was taken so that those farmers were not approached for the same survey by the researcher. No other attempt was made to contact participants, and no compensation of any kind was offered to responders. The initial telephone approach was excluded to protect the personal data of the target group, despite the positive effect on the response rate [51,54].

### 2.3. Statistical Analysis

Data were analysed using IBM SPSS statistics for Windows, version 27 [55]. Descriptive statistics were performed for demographic characteristics and to describe responses concerning knowledge. All records were examined for being within the expected values, for missing values and for any other errors. Outliers were identified, verified that they were not attributed to errors and kept for statistical analysis. Differences between participants’ categories and between applied practices for safe manure management were assessed using the Chi-Square test.

The distribution of the “Total knowledge score” was checked for normality using the Kolmogorov–Smirnov and the Shapiro–Wilk tests. The tests indicated that there was a deviation from normality (*p* < 0.001). It was not possible to normalise the distribution with data transformation. As a consequence, to compare differences in knowledge scores between demographic groups, the Mann–Whitney U Test was used as an alternative to the Independent Samples *t*-test. The farm size, education level, experience, and farmers’ age were dichotomised for the analysis. Farm size was divided into farms with equal or less than 250 bovines and those with more than 250 animals because the bigger farms, typically, are managed by professional farmers, whereas smaller farms are managed by farmers who practice farming as a secondary occupation. The education level, farmers’ age, and experience, which are reflected in the years in the occupation, were divided at the median value. The Kruskal–Wallis Test was used to compare the variance of the total score between the districts. To further investigate whether various demographic characteristics are determinants of participants’ knowledge scores, multiple linear regression analyses were performed with the knowledge score as the dependent variable and the farming experience, education level, herd size and farming purpose as independent variables. Additionally, the variable of “Farming Purposes” was used after merging the “dairy purpose” with “a mixed production purpose” (dairy and beef) because all mixed-purpose farms in Cyprus have cattle that belong to dairy breeds, and they are raised as an appendant activity.

To compare the knowledge of groups with different beliefs and attitudes towards good farming practices, the Mann–Whitney U-Test was used. To perform the test, Likert scale questions were dichotomised by merging “Not at all Important” with “Slightly Important” and “Important” with “Very Important” answers [56,57]. The “Not sure/No opinion/Don’t know” answers were treated as blanks.

For all analyses, the *a*-value was determined at 0.05, and any *p*-value less than 0.05 was considered significant. Missing data were considered blanks and were excluded pairwise from analyses.

## 3. Results

### 3.1. Description of the Responders

Of the 353 farmers approached, 105 returned completed questionnaires, representing a response rate of 30%. The general characteristics of the participants are presented in Table 1. Most of the participants were in the age group of 55–64 years. Most of the farmers had long farming experience, usually more than 30 years and had finished secondary education (Lyceum). Responders were predominantly from the Lefkosia district, following Paphos and Larnaka. The majority had a mixed farming purpose (dairy and beef). Farms having up to 250 animals predominated.

### 3.2. Description of Practices for Manure Management

About 85% of participants used manure as a fertiliser. Almost 70% of those using it are in dry form. Nearly one-third of stored manure is in piles at a dedicated area of the farm, usually with cement floors. The majority of participants stored manure for more than three months before dispatching it to be used as a fertiliser without any further treatment (Table 2).

### 3.3. Evaluation of Knowledge

The mean knowledge score achieved by participants at the four types of questions as well as the total score, are presented in Table 3. For most questions, the score was above 50% of the maximum, apart from the question concerning the transmission of diseases and other hazards from contaminated manure, which was 43.5% of the maximum score. The score did not approach full marks in any of the questions, with the highest achieved being 68.5% of the maximum in the question about good farming practices. A detailed breakthrough of the answers to each of the questions is presented in Appendix B (Table A1).

### 3.4. Participant Characteristics in Relation to Knowledge Level and Applied Practices

Comparing the median scores of farmers with different characteristics revealed that participants managing larger farms, participants managing dairy or dairy with beef cattle farms, and participants with an educational level of lyceum and above had significantly higher scores. Farmers from the Paphos district scored significantly lower than other districts. A detailed comparison of scores obtained by farmers with different characteristics is presented in Appendix B (Table A2). Further analysis to investigate whether the farming experience, education level, herd size and farming purpose were determinants of the knowledge score was carried out. The results of the multiple linear regression analysis suggested that the education level and the farming purpose could significantly influence the score (Table 4). In particular, the higher the education level, the higher was the score. Likewise, farmers who kept only animals of the local breed had lower scores than farmers who kept animals for beef purposes only, and the latter group scored lower than those who kept dairy animals or animals for mixed, dairy and beef purposes. The age was excluded from the analysis because of significant collinearity with farming experience. The district of the farm was also excluded because of the presence of outliers. The final model, which included only the two significant predictor variables, indicated that it could explain 39.3% of the variance of knowledge score [F(2, 92) = 31.46, *p* < 0.0005, R^2^ = 0.39]. Both variables, education level (B = 1.66, *p* = 0.03) and farming purpose (B = −4.287, *p* < 0.0005), contributed significantly to the model.

Farmers who had positive attitudes and beliefs towards specific good farming and manure management practices had a higher knowledge level about proper manure management (Table 5). In particular, farmers who considered vaccination, mice control programmes, animal welfare and proper manure management important for farming achieved significantly higher scores than those who did not consider these issues important. Regarding treatment methods, those who did not identify certain manure methods that increase safety as important achieved significantly lower scores. Specifically, the groups which considered not important the storage in piles (with mixing to promote fermentation), the storage in lagoons or tanks for at least six months, composting and air-drying under the sun had significantly lower scores than other groups. The scores of groups with different beliefs about pasteurisation, transformation into biogas, and addition of chemicals as methods for reducing manure risks did not show any significant differences.

## 4. Discussion

The results of the study characterise the level of knowledge about safe manure management and the practices used for that purpose by bovine farmers in Cyprus. The questionnaire analysis showed that despite the certain level of knowledge among farmers about safe manure management, they underestimate the importance of manure in spreading common pathogens, and more than half do not realise manure’s role in spreading serious infectious diseases. The results indicated that the farming purpose and the educational level of farmers are associated with better knowledge of the risks arising from improper manure management, whereas farmers’ age and total years in the occupation do not play a significant role. There are also geographical areas where the level of knowledge is lower than others. Regarding current practices, the use of manure as a fertiliser for crops is prominent. Typically, bovine farmers store their manure before dispatch. The fact that the manure is stored uncovered in a dedicated area with a cement floor, where it remains for more than three months until it becomes dehydrated, diminishes its pathogen levels.

Farmers’ knowledge about health risks from manure is extremely important to avoid practices that may spread diseases [59]. It appears that most farmers have a certain degree of knowledge about the risks arising from improper manure management. Šūmane et al. [60] indicated the importance for livestock farmers of having a certain knowledge level on new developments to maintain sustainability and resilience. Although there is no benchmark to compare knowledge on manure management, it would be beneficial if farmers were more aware of possible risks and the relevant mitigation measures. Responses to the questionnaire revealed some misconceptions about farmers in relation to the risks. In particular, their knowledge about diseases spreading through manure is not adequate considering that they achieved, on average, less than half of the maximum overall score. Farmers underestimate the importance of manure in spreading common pathogens, such as those responsible for enteritis and intestinal parasites, but also more than half do not realise manure’s role in spreading serious zoonotic and other infectious diseases that might have devastating effects on the sector, such as foot and mouth disease. This agrees with the finding of Eltayb et al. [61], who demonstrated that only half of the farmers in their study had knowledge of zoonotic diseases. Although these diseases are considered exotic in Cyprus, appropriate biosecurity measures should be in place to prevent the occurrence or diminish the consequences of any outbreaks [62].

There are certain specific practices that may reduce manure’s pathogen loads [10]. The detailed analysis of the results showed that farmers had some difficulties in correctly identifying practices that promote the reduction of pathogen loads, namely vaccination programmes, waiting for manure to become digested before using it as a fertiliser or soil improver and maintaining a high level of animal welfare. Regarding knowledge about manure treatment methods, almost two-thirds of participants were aware that these contribute to the reduction of pathogens. The effectiveness of manure’s storage in lagoons or tanks for a period of time seems to be underestimated. Some confusion also exists on matters such as the timing for manure use in crops. Overall, farmers have some knowledge gaps about proper safety practices in all stages of production, from the farming practices to storage, treatment and use of manure. The above results align with the conclusions of Ström et al. [59], who also pointed out that the lack of knowledge is a key element hindering safe manure management. In general, the knowledge about proper manure management must be reinforced, giving emphasis to areas of weakness, such as the transmission of certain diseases and good farming practices.

Implemented practices for manure management are extremely important for safety [10]. Among participants, the application of certain practices for manure management is prominent. As a rule, manure produced in bovine farms is used on land as a fertiliser. Very few farmers submit it to composting or biogas transformation at an approved plant prior to its use. Because these treatments reduce pathogens, especially when equipped with a pasteurisation unit [63,64], the importance of applying other good management practices becomes obvious. Unfortunately, only a negligible number of farmers perceived the prospect of utilising manure as an energy source in biogas plants. This supports the findings of Meyer et al. [65], who reported only one out of 394 farms surveyed used biogas transformation before manure was used as a fertiliser. This might be attributed to insufficient information, absence or deficient capacity of biogas plants nearby [66,67]. Promoting biogas and composting plants in areas with farming activities provides a sustainable alternative to managing manure, which enhances safety while increasing profitability in an environmentally friendly manner [66,68]. Possible financial constraints, lack of guidance, state aid, or infrastructures, insufficient legislation or other reasons may explain why farmers have restricted access to those plants and justify their approach to manure management. Therefore, more research is needed to provide insights for better utilisation of these methods, including the investigation of the above factors and their influence on the development of the methods.

Current practices before manure use on land, though not optimal, contribute to the sanitisation of the material. The majority of farmers store manure on the farm before use. The storage is usually done in a dedicated area, often with cement floors or in watertight tanks, which prevent contamination of the environment. This practice deviates from previous studies [7,69] that place large lagoons as the most common system of storage. This could be attributed to the implementation of the national environmental rules, which determine the accepted methods for in-farm storage [70]. The storage period varies, but as a rule, it exceeds three months. Microbiological processes during the prolonged storage period enable the aging or maturation of manure and the reduction of pathogens. The dehydration occurring during the process eliminates the available water for bacteria proliferation [9,10]. In conclusion, applied practices, which are driven mostly by the national environmental legislation, reduce the risks from pathogens and other contaminants, although they do not ultimately ensure safety for the environment, human and animal health.

Participants with specific characteristics achieved higher scores on knowledge about safe manure management. These include the size of the farm, farming purpose, area of the farm and the participants’ educational level. Wang et al. [71] and Singh et al. [72] indicated the relevance of education to the level of knowledge for the use of pesticides in farming. The multiple regression analysis has shown that the farming purpose and the farmer’s educational level have a significant effect on knowledge scores. The fact that a farmer’s age and years of experience do not seem to play a significant role suggests that life and farming experiences are not so important for the knowledge level as the specialised training. This points out the necessity of providing training to all farmers, emphasizing the importance of safe manure management to human and animal health and the environment. These findings also agree with Singh et al. [72], who evaluated farmers’ knowledge of zoonotic diseases; however, they indicated that there was a negative association between the age of participants and knowledge. Hundal et al. [73] concluded that age, education level and farm size are not significant determinants of knowledge about zoonotic risks. However, Sumane et al. [60] supported that experiences and own experimentation should not be underestimated. Usually, the farm size is linked to the level of professionalism. Farmers with smaller farms might practice farming as a second occupation, whereas bigger farms are usually operated by professional farmers, and the farming system is more intensive. This is particularly true about very small farms, usually with less than six animals. Farmers who receive their sole income from farming may be more concerned about safety risks and try to educate themselves more. The farming purpose also correlates with professionalism. Professional farmers usually manage dairy farms. Often, they keep male calves for beef production. Therefore, they have a mixed farming purpose. In Cyprus, rearing cattle for beef production, as a sole activity, is not as profitable as dairy farming. Furthermore, only dairy breeds are available for beef production. Consequently, it is common for exclusively beef cattle farmers to practise in parallel to other occupations. The same applies to farmers of bovines belonging to the local breed, which is the reason that rearing takes place in villages by non-professional farmers. A lot of these farms are encountered in the Paphos district, where the majority of bovine farms have few animals. These might be reasons for the poor results of this district compared with all other districts. Saha et al. [74] indicated the negative influence of small herd size on the knowledge level of animal rearing practices; however, there are studies supporting that herd size is not affecting knowledge [73]. In addition, the absence of any bovine farmers’ association in the area may deprive farmers of the opportunity for organised training [60]. The limited number of demographic factors identified as contributing to the knowledge level and the percentage of variance explained by the regression models indicate that further research is needed to detect other factors affecting the knowledge and practices used, including sources of information available to farmers, whether they have previously received relevant training, the relevance of their studies and others.

In our study, most farmers who considered good farming practices and some manure treatment methods not important achieved significantly lower scores than those who believed that these good farming practices are important for reducing the risk arising from manure. Although it is not clear what reasons shaped these attitudes, insufficient knowledge could be one of them, as Musallam et al. [75] demonstrated. The absence of significant differences between groups with different beliefs about the treatment methods of pasteurization and transformation into biogas is noteworthy. These methods are considered among the most effective against pathogens of manure, especially in combination are very efficient for both biological and chemical hazards effectiveness [32,37,38,39,40]. The absence of differences in scores between the groups reveals that farmers with both high and low scores might have equally incorrect beliefs and unveils the confusion about these methods. This indicates the lack of relevant information to farmers and the gap of knowledge that exists about the best available technics for controlling the hazards found in manure before using it as a fertiliser. Therefore, educating those farmers might contribute to the change in beliefs and attitudes and promote the application of safer farming practices [76].

To reinforce the attenuation of enzootic diseases and prevent new diseases, the parties involved in farmers’ training need to inform farmers about the health and environmental risks associated with manure and the methods that can be applied to mitigate those risks. Competent authorities, private veterinarians and farmers’ organisations can contribute to this by organising training sessions, awareness campaigns, preparing educational material and informing farmers accordingly [9,65,77]. Equally important is farmers’ networking, exchange of experiences and developing of practical skills [60]. The outcome of this research indicates that training is needed for all farmers. It also identifies the areas where training should focus on, the farmers’ groups that would particularly benefit, and provides guidance for appropriate interventions. Furthermore, the regulatory authorities may consider the amendment of the legislative framework for encouraging the application of manure treatment methods, setting microbiological criteria before manure use on land, introducing subsidies to develop biogas and composting plants and providing advice for running those facilities.

The main strength of this research is that it is the first attempt to gather information about farmers’ knowledge of safe manure management. The exclusion of phone and personal contact with participants reduced the potential bias of study results. The tool of the survey, which was a printed questionnaire, was more user-friendly to people unfamiliar with modern technologies. Although the questionnaire was completely anonymous, some farmers might be reluctant to participate for fear of spotting them as having limited knowledge or not implementing proper practices for manure management. Therefore, the average knowledge score and the safety of applied practices could be overestimated. This might contribute to the dichotomised possible answers (right/wrong, yes/no). Although this facilitated the completion of the questionnaire, it could lead to bias because the responders had a 50% chance of selecting the correct choice, even if they did not really know the answer. In addition, the information that the questionnaire collected for the applied farmer practices regarding the storage of manure could have been more detailed. A potential limitation of the study is the lack of information on the precise number of animals on each farm, which restricted the ability to examine differences in subgroups of herd sizes, especially in the smaller herd sizes (up to 250 animals). A further limitation is that the time the survey was executed coincided with a busy period for farmers, which might deter them from participation. Nevertheless, the 30% response rate was higher than that of Benjamin et al. [57], who followed the same approach but achieved a 26% response rate. Other studies reported slightly higher response rates; however, they encourage participation by providing incentives [78] or by telephone contacts [51,54].

Because of the distinctive circumstances of Cyprus, such as the use of dairy breeds for beef production, climatic conditions, restricted access to manure processing facilities, local farmers’ organisation schemes and the restricted island area, the generalisation of the results should be done with caution to settings and locations with different circumstances. Nevertheless, this study could be useful as baseline information for further similar research at other locations.

## 5. Conclusions

Despite the certain level of knowledge among farmers about public and animal health risks associated with manure management, their knowledge should be reinforced. Educating farmers may improve current management practices, which are not optimal for ensuring the safe use of manure and the sustainability of the sector. By promoting safe manure management, the sector intensifies its sustainability and public acceptance because of the lower impact on the environment and human and animal health. Authorities and other stakeholders should intervene to improve knowledge and promote safe practices. Training and education of farmers as well as promoting appropriate manure treatment methods, should be the core elements of these interventions. The farming purpose and educational level, as the most important factors associated with a high knowledge level could guide the prioritisation of training interventions.

## Figures and Tables

**Table 1 vetsci-10-00293-t001:** General characteristics of participants.

Variable	Category	n	%	*p*-Value ^1^
Age (years)	18–24	102	1	<0.001
	25–34		6.9	
	35–44		23.5	
	45–54		24.5	
	55–64		33.3	
	>65		10.8	
Farming experience (years)	<10	100	13	<0.001
	10–19		11	
	20–29		19	
	30–39		27	
	40–49		26	
	>50		4	
Education	Primary (Elementary)	105	6.7	<0.001
	Gymnasium		20	
	Lyceum		53.3	
	University		19	
	None of the above		1	
District of the farm	Lefkosia	103	30.1	<0.001
	Ammochostos		12.6	
	Larnaka		22.3	
	Lemesos		6.8	
	Paphos		28.2	
Herd size (number of animals)	1–6	105	6.7	<0.001
	7–250		58.1	
	251–500		30.5	
	501–750		1.9	
	751–1000		1.9	
	>1000		1	
Farming purpose	Dairy (exclusively)	104	15.4	<0.001
	Beef (exclusively)		2.9	
	Mixed		62.5	
	Local breed		19.2	

n—Number of total responses in each category; %—Percentage of the total in each category. ^1^ The *p*-value was obtained by analysing the data using the Chi-Square Goodness of Fit test.

**Table 2 vetsci-10-00293-t002:** Practices applied by participants for manure management.

Question	n	Possible Answers	Number of Answers	%	*p*-Value ^1^
How do you use manure produced on your farm?	94	As a Waste	5	5.3	<0.001
		As a Fertiliser	80	85.1	
		As an Energy Source	1	1.1	
		Any other uses	8	8.5	
How often do you send manure from your farm for final use?	102	Every week	12	11.8	<0.001
		Every month	12	11.8	
		Every quarter	11	10.8	
		More than three months	67	65.7	
In case the manure is used as a fertiliser, in which form is it used?	101	Liquid form	4	4	<0.001
		Air dried form	70	69.3	
		Both liquid and air dried	27	26.7	
How do you store manure at your farm before you send it to another place?	130	In a watertight tank	28	21.5	<0.001
		In uncovered piles	32	24.6	
		In covered piles	6	4.6	
		In a dedicated area with cement floor	37	28.5	
		In a place with earthen ground	15	11.5	
		In any place that is convenient	5	3.8	
		I don’t store it at all	7	5.4	
To where do you dispatch the manure produced on your farm?	103	To be used directly as a fertiliser in the fields	88	85.4	<0.001
		To a biogas plant	3	2.9	
		To become fertiliser in a composting plant	10	9.7	
		To be disposed of as a waste	2	1.9	
		To be incinerated	0	0	

n—Number of total responses in each question; %—Percentage of the responders who chose each answer. ^1^ The *p*-value was obtained by analysing the data using Chi-Square Goodness of Fit test.

**Table 3 vetsci-10-00293-t003:** Participants’ knowledge concerning safe manure management ^1^.

Question	Mean Score	SD ^2^	% of Maximum Score ^3^
Which actions do you believe help in reducing the risk of transmission of diseases to animals or humans from the improper manure management?	4.1	0.87	68.5
2. Which diseases/hazards can be transported to other areas via manure?	4.8	2.52	43.5
3. Which methods of manure treatment reduce the risks to animal and human health?	5.1	2.54	64.3
4. Right or wrong statements	4.1	1.69	67.8
Total score	18.1	5.92	58.4

%—Percentage. ^1^ Results are based on 101 participants. ^2^ SD—standard deviation. ^3^ Maximum possible score: for question 1, it was 6 points; for question 2, it was 11 points; for question 3, it was 8 points; and for question 4, it was 6 points. The maximum total score was 31.

**Table 4 vetsci-10-00293-t004:** Determinants of farmers’ knowledge for safety manure management.

	Total Knowledge Score		
	β	95% CI	*p* *
Farming experience (years)	0.037	−0.07–0.10	0.69
Education level	0.192	0.06–3.47	0.043
Herd size (number of animals)	0.158	−0.17–2.48	0.087
Farming purpose	−0.497	−5.2–−2.36	<0.0005
R^2^	0.429	*p* **	<0.0005
Adjusted R^2^	0.402	F	16.131

The standard method for multiple linear regression was used. β—Standardised beta coefficients; CI—Confidence Interval; *p* *—significance of coefficients, *p* **—significance of the analysis.

**Table 5 vetsci-10-00293-t005:** Comparison of knowledge scores of farmers with different attitudes and beliefs regarding safe manure management and farming practices.

Question about Attitudes and Beliefs	Statement	n	Median Score ^1^	*p*-Value ^2^	Effect ^3^ Size (*r*)
How important do you believe is the treatment of manure to reduce risks to animal or human health with each of the following methods before it can be used as a fertiliser?					
Storage in piles for at least 40 days with mixing for ventilation	Not important	26	16 (14, 21.5)	0.015	0.26
	Important	61	22 (19, 23)		
Storage in lagoons or tanks or tanks for 6 months	Not important	33	20 (16, 23)	0.016	0.29
	Important	35	22 (20, 24)		
Pasteurisation (70 °C for an hour)	Not important	21	21 (16, 23.5)	0.297	
	Important	37	22 (20, 23)		
Air-drying under the sun	Not important	14	16.5 (13.75, 23)	0.047	0.21
	Important	72	21 (19, 23)		
Composting	Not important	11	20 (14, 22)	0.049	0.23
	Important	62	21.5 (19, 23)		
Transformation into biogas	Not important	14	19.5 (15.5, 23.25)	0.097	
	Important	56	22 (20, 23)		
Addition of lime or other chemicals for sanitization purposes	Not important	18	21 (16, 23)	0.08	
	Important	51	22 (20, 23)		
How important do you consider the following actions for your farm?					
Vaccination program	Not important	17	16 (13,18.5)	0.001	0.33
	Important	80	21 (18,23)		
Mice control	Not important	11	11 (6, 14)	<0.0005	0.43
	Important	87	21 (18, 23)		
Increase in milk production	Not important	7	15 (12, 22)	0.042	0.22
	Important	80	21 (19, 23)		
Genetic improvement	Not important	7	12 (9, 16)	0.001	0.35
	Important	81	21 (18.5, 23)		
Production of feed	Not important	13	20 (14, 23.5)	0.378	
	Important	77	21 (18, 23)		
Animal welfare	Not important	6	13 (10.5, 17.25)	0.005	0.29
	Important	85	21 (17.5, 23)		
Manure management	Not important	12	16 (12.5, 20.75)	0.014	0.25
	Important	80	21 (18, 23)		

n—Number of total responses who chose the particular answer. ^1^ The score values are presented as median (interquartile score). ^2^ The *p*-value was obtained by the Mann–Whitney U Test. ^3^ The effect size refers to the Mann–Whitney U Test and is evaluated using Cohen [58] criteria of 0.1 = small effect, 0.3 = medium effect, 0.5 = large effect.

## Data Availability

Data will be available upon request by interested parties.

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
