# Peer review of "Knowledge and Practices of Cypriot Bovine Farmers towards Effective and Safe Manure Management"

_vetsci, 2023, doi:10.3390/vetsci10040293_

Round 1
Reviewer 1 Report
in the attached file

Reviewer 2 Report
General comments:
This study aimed to evaluate the practices of manure management in confined bovine farms at Cyprus using a national survey. Overall, this study is well designed and structured, and clearly presented. One point that I suggest to clarify is if some farms process a solid-liquid separation. This aspect can be important mainly when pits/thanks are not used to store manure. Also, for those farms drying manure (without liquid/solid separation), the manure is stored in a permeable floor or not; I.e., what is the role of unprocessed liquid infiltration on lands. Please define “dried” manure (air dried?). I believe that some discussion can be done about these issues to well characterize the current procedures. What is the current national legislation for manure processing?
Specific comments:
L47-48: Please merge this information with L58-60.
L52- Please remove the reference [6].
L91-95: There are national legislation? If yes, please report the main measures (in introduction or discussion sections).
L119-120: All farms with confined or semi-confined animals? For putative “semi-confined systems”, what is the animal density at grazing; there are rotational management? I summary, I suggest a more deeply characterization of the farms.
L245: Can this work establish a relationship between herd size and manure management?
